# Artificial Intelligence for Automated DWI/FLAIR Mismatch Assessment on Magnetic Resonance Imaging in Stroke: A Systematic Review

**DOI:** 10.3390/diagnostics13122111

**Published:** 2023-06-19

**Authors:** Cecilie Mørck Offersen, Jens Sørensen, Kaining Sheng, Jonathan Frederik Carlsen, Annika Reynberg Langkilde, Akshay Pai, Thomas Clement Truelsen, Michael Bachmann Nielsen

**Affiliations:** 1Department of Radiology, Copenhagen University Hospital Rigshospitalet, 2100 Copenhagen, Denmark; jens.sorensen@sund.ku.dk (J.S.); kaining.sheng@regionh.dk (K.S.); jonathan.frederik.carlsen@regionh.dk (J.F.C.); annika.langkilde@regionh.dk (A.R.L.); ap@cerebriu.com (A.P.); mbn@dadlnet.dk (M.B.N.); 2Department of Clinical Medicine, University of Copenhagen, 2200 Copenhagen, Denmark; thomas.clement.truelsen@regionh.dk; 3Cerebriu A/S, 1127 Copenhagen, Denmark; 4Department of Neurology, Copenhagen University Hospital Rigshospitalet, 2100 Copenhagen, Denmark

**Keywords:** MR DWI/FLAIR mismatch, artificial intelligence, machine learning, wake-up stroke

## Abstract

We conducted this Systematic Review to create an overview of the currently existing Artificial Intelligence (AI) methods for Magnetic Resonance Diffusion-Weighted Imaging (DWI)/Fluid-Attenuated Inversion Recovery (FLAIR)—mismatch assessment and to determine how well DWI/FLAIR mismatch algorithms perform compared to domain experts. We searched PubMed Medline, Ovid Embase, Scopus, Web of Science, Cochrane, and IEEE Xplore literature databases for relevant studies published between 1 January 2017 and 20 November 2022, following the Preferred Reporting Items for Systematic Reviews and Meta-Analyses guidelines. We assessed the included studies using the Quality Assessment of Diagnostic Accuracy Studies 2 tool. Five studies fit the scope of this review. The area under the curve ranged from 0.74 to 0.90. The sensitivity and specificity ranged from 0.70 to 0.85 and 0.74 to 0.84, respectively. Negative predictive value, positive predictive value, and accuracy ranged from 0.55 to 0.82, 0.74 to 0.91, and 0.73 to 0.83, respectively. In a binary classification of ±4.5 h from stroke onset, the surveyed AI methods performed equivalent to or even better than domain experts. However, using the relation between time since stroke onset (TSS) and increasing visibility of FLAIR hyperintensity lesions is not recommended for the determination of TSS within the first 4.5 h. An AI algorithm on DWI/FLAIR mismatch assessment focused on treatment eligibility, outcome prediction, and consideration of patient-specific data could potentially increase the proportion of stroke patients with unknown onset who could be treated with thrombolysis.

## 1. Introduction

Magnetic resonance imaging (MRI) is the primary imaging modality for stroke detection and classification in patients with an unknown onset and wake-up stroke (WUS), using Diffusion-weighted imaging (DWI) and T2-Weighted Fluid-Attenuated Inversion Recovery (FLAIR) sequences [1]. DWI sequences are useful for the detection of early signs of infarction and can be used to outroot stroke mimics [2,3]. The visibility of hyperintense FLAIR sequence lesions compatible with acute infarction increases with time since stroke onset (TSS). However, the intensity of FLAIR hyperintensity lesions cannot provide an exact time of stroke onset [4,5,6,7,8]. 

Based on results from randomized clinical trials, alteplase or recombinant tissue plasminogen activator (rtPA) can be administered up to 4.5 h after stroke symptoms onset, as the risk of hemorrhage and poor outcomes increases with time [6,7,8,9]. For patients with unknown stroke onset, which accounts for 20% to 27% of all ischemic strokes, the DWI/FLAIR lesion mismatch assessment is used to assess whether the patient can be treated with rtPA [8,10,11]. Presently, only Wake-up stroke patients/patients with unknown TSS without visible FLAIR lesions and within 4.5 h from recognition of stroke symptoms are considered eligible for treatment [8,9]. 

Over recent years, the number of artificial intelligence (AI) algorithms applicable for radiological purposes has increased markedly [12,13,14]. Multiple studies have shown the ability of algorithms to detect whether ischemic lesions are present on DWI and FLAIR sequences with deep learning (DL) algorithms [15,16]. 

Field experts, e.g., neuroradiologists, perform the current clinically used binary mismatch assessment, but it has been shown to have poor inter- and intra-observer agreement [6,7,8]. An automated assessment of DWI/FLAIR mismatch could create a more standardized method and assist the stroke progression analysis and treatment selection process. 

We conducted this Systematic Review to discover the currently existing AI algorithms for automated DWI/FLAIR mismatch assessment and determine how well DWI/FLAIR mismatch algorithms perform as compared to experienced radiologists and neurologists.

## 2. Materials and Methods

This systematic review was conducted in accordance with the Preferred Reporting Items for Systematic Reviews and Meta-Analyses (PRISMA) statement [17]. The study protocol was registered in the Prospective Register of Systematic Reviews (PROSPERO) under number CRD42022377938 during the research process.

### 2.1. Literature Search

Literature was searched between 26 October 2022, and 20 November 2022, in PubMed MEDLINE, Ovid Embase, IEEE Xplore, Web of Science, Elsevier Scopus, and Cochrane. We searched articles published between 1 January 2017, and 20 November 2022, within the clinical and technical scope of the review.

The preliminary search string was done in PubMed MEDLINE. Afterward, it was converted and edited with the polyglot tool Systematic Review-accelerator (Bond University, Australia) [18]. This made the search strings suitable for the rest of the databases. Search strings for each database can be found in Appendix A.

### 2.2. Study Selection

The literature was extracted and collected for the reference tool Endnote 20 (Clarivate, London, UK). Duplicates were removed with Endnote’s internal duplicate search function.

The references were exported to Covidence (Melbourne, Australia) to screen and review articles. The internal duplicate tool in Covidence removed duplicates not caught by Endnote 20. 

Only studies focusing on machine learning methods to automatically identify DWI and FLAIR lesions on structural brain MRI were included. The studies had to be peer-reviewed and in English. Editorials, case series, letters, conference proceedings, reviews, and inaccessible papers were also excluded. The main inclusion and exclusion criteria for the screening process are shown in Table 1.

The two reviewers, a medical doctor (C.M.O.) and a medical student (J.S.), independently screened all records based on titles and abstracts, followed by a full-text review of the potentially relevant papers for final inclusion. Any conflicts during the screening process were resolved by consulting a 3rd and 4th reviewer (K.S. and J.F.C.).

### 2.3. Data Extraction and Analysis

Three reviewers (J.S., C.M.O., and K.S.) extracted data from the included articles. The extracted study characteristics composed of the following: The number of patients, size of training and test data set, population demographics, National Institutes of Health Stroke Scale (NIHSS) score upon admission, time from onset to MRI, and the percentage of the dataset with an onset to MRI within 4.5 h from symptom onset. In addition, scanner characteristics, the type of scanner, and the used artificial intelligence methods were extracted. Performance data for the algorithms and used comparator was extracted and composed of; Area under the curve (AUC), Sensitivity (Sens), Specificity (Spec), Accuracy (Acc), Precision or positive predictive value (PPV), and negative predictive value (NPV).

If multiple performance data were mentioned for different algorithms, only the performance data from the best algorithm was selected unless otherwise stated.

### 2.4. Quality Assessment

The two reviewers used the questionnaire Quality Assessment of Diagnostic Accuracy Studies 2 (QUADAS-2) tool to perform the quality assessment of the included studies [19]. The QUADAS-2 assess the risk of bias and concern for applicability in the domains of patient selection, index test, reference standard, and flow and timing. The domains were categorized as having a high, unclear, or low risk of bias or concern for applicability.

## 3. Results

### 3.1. Study Selection

The literature search resulted in 3273 articles. Duplicates and conference abstracts accounted for 1595. The total number of titles and abstracts to screen was then 1678. Eighteen articles were selected relevant for full-text assessment. In five articles, the AI algorithm did not perform a DWI-FLAIR mismatch assessment. One was not in English. Two had outcomes not relevant for this study, i.e., a focus on prediction accuracy for upper extremity motor outcomes 90 days post-stroke and a focus on detecting FLAIR and DWI lesions independently and not as a mismatch [20,21]. One article was a preprint of an already-included article. Ultimately, this led to a total number of five articles included in the review. This study’s inclusion process is presented in Figure 1.

### 3.2. Study Characteristics 

All five included studies were retrospective. They used image data from different hospitals or university databases from the period between 2011 and 2021. Study characteristics and scanner specifications are summarized in Table 2. 

The study population sizes varied from 268 to 587 patients diagnosed with stroke. The study conducted by Polson et al. had two study populations, one internal and one external [24]. The external population was the same as the one used in the study by Lee et al. [25]. Therefore, the internal study population from Polson et al. was chosen for reporting in this review. All five studies divided their dataset into training populations and test populations. In Lee et al., Polson et al., and Zhang et al. the distribution of patients in the two groups was very similar, ranging from 16% to 19% [23,24,25]. Zhu et al. and Jiang et al. had a much higher proportion of the test population; the distributions were 30% and 40%, respectively [22,26]. 

The median population age covering all five studies ranged from 63 to 70 years. The maximal difference in NIHSS scores upon admission between training and test populations in the included articles was minor. Polson et al. and Zhang et al. had a higher NIHSS score upon admission in the training population compared to the test population [23,24]. Lee et al. and Jiang et al. had a higher score in the test population [25,26]. Zhu et al. only reported an average score for the whole study population [22]. 

Zhu et al. had a very high average time from onset to MRI of 10.3 h [22]. The median time from stroke onset to a performed MRI in the four other studies ranged from 3.5 h to 4.5 h across training and test populations [23,24,25,26]. Polson et al., Jiang et al., and Zhu et al. stated the percentage of patients who had a stroke onset time to MRI within 4.5 h [22,24,26]. The variation across training and test population was 64.55% to 50%. 

Four out of five studies used stroke neuroradiologists or neurologists for mismatch assessment as the comparator for their AI algorithm [22,23,24,25]. Polson et al. and Zhang et al. used the independent mismatch assessments of three neuroradiologists and aggregated their results [23,24]. Lee et al. used two neurologists’ mismatch assessments, potential disagreements were resolved, and a consensus was obtained [25]. Zhu et al. used an unspecified number of neuroradiologists for the mismatch assessment [22]. Jiang et al. did not have a human comparator in the study, but due to the scope of this review, it was relevant for inclusion [26].

### 3.3. Automated Assessment of Time since Stroke

Lee et al., Jiang et al., and Zhu et al. used a DL algorithm in conjunction with conventional machine learning (ML) algorithms in a two-step model for the assessment of TSS [22,25,26]. The first step for all three studies was the segmentation of the stroke lesion on the DWI sequence to determine the region of interest (ROI) and volume of interest (VOI) using DL algorithms derived from the U-net architecture. The specific segmentation methods are seen in Table 2. 

After the segmentation, all three studies used different ML techniques for the TSS classification. All three studies tested and compared three to seven various ML models to find the best-performing ML model. In Lee et al. and Jiang et al., the best-performing ML models were Random Forest and a support vector machine with a radial kernel (svmRadial), respectively [25,26]. Zhu et al. used an ensemble approach with one ML model integrating the outputs of the top five best-performing ML algorithms to produce a final prediction [22]. Polson et al. and Zhang et al. only used an end-to-end convolutional neural networks model (CNN) for stroke segmentation and mismatch assessment [23,24]. Both studies trained the CNN to segment the lesion on undefined sequences and afterward trained it to classify TSS using one unified architecture.

### 3.4. Study Performance 

For this review, the performance measures for the AI models were registered, and missing values were calculated from the existing data when possible. The best performance metrics and preferentially from out-of-distribution external validation datasets are the performance results for the TSS classification reported in Table 3. 

The area under the curve (AUC) was reported in all five studies. It ranged from 0.74 to 0.90 for the AI models. The AUC was not reported for the comparators, as these assessments were dichotomous. The sensitivity, also known as the true positive rate, and the specificity, also known as the true negative rate, were reported in all five studies. It ranged from 0.70 to 0.82 and from 0.74 to 0.84, respectively. Evaluation of the best performances showed that the specificity was a little higher than the sensitivity in all five studies. Results are shown in Table 3.

For Lee et al., Jiang et al., Polson et al., and Zhu et al., the positive predictive value (PPV) and negative predictive value (NPV) were reported or calculated, ranging from 0.74 to 0.91 and 0.65 to 0.80, respectively. In the same four studies, the accuracy ranged from 0.73 to 0.83 [22,24,25,26]. In Zhu et al., precision, also known as PPV, was registered for the machine learning methods, but neither precision nor PPV was registered for the comparators. PPV was therefore calculated for both the AI method and the comparators in order to align with the other studies. 

Performance measures for the comparators were obtained in all studies except for Jiang et al. The sensitivity and specificity ranged from 0.49 to 0.82 and 0.59 to 0.91, respectively. The accuracy ranged from 0.61 to 0.74 [22,23,24,25]. For Zhu et al., Polson et al., and Lee et al., PPV and NPV for the comparators were either reported or calculated. It ranged from 0.73 to 0.89 and 0.55 to 0.72, respectively [22,24,25]. 

These results suggest that AI sensitivity, NPV, and accuracy in the determination of TSS were better than for the radiologists’ but the AI specificity and PPV were comparable to that of the radiologists’ in a binary classification of TSS ±4.5 h. 

### 3.5. Quality Assessment

The Quality Assessment of Diagnostic Accuracy Studies 2 tool was applied to all included studies of this review. The results of the risk of bias/concern for applicability analysis are presented in Table 4.

Cohort sizes were reasonably large in all five included studies and comparable to the cohort sizes in studies included in another systematic review with a focus on the potential of automated segmentation of stroke lesions in MRI images [27]. 

Sampled study cohorts, description of inclusion and exclusion criteria, and processing of the images for optimization of the AI algorithm performances were assessed.

The studies description of the comparator process and methods used for the segmentations also affected the evaluation.

None of the studies were assessed with a low risk of bias and concern for applicability in all domains. 

No meta-analysis was conducted due to the small number of included articles.

## 4. Discussion

The currently existing AI-based DWI/FLAIR mismatch assessment uses FLAIR lesion visibility as a surrogate marker of TSS with a binary decision, i.e., ±4.5 h since stroke onset. Because of the non-trivial correlation between FLAIR changes and TSS, it is not recommended that FLAIR lesion visibility is used for the determination of time since stroke onset within the first 4.5 h [5]. However, an AI-assisted DWI/FLAIR mismatch assessment could be useful in patients of unknown onset time/wake-up stroke if a mismatch is identified and the time from recognized symptoms is less than 4.5 h. Then the patient could receive rtPA treatment. 

### 4.1. Artificial Intelligence vs. Human Readings

Ebinger et al. have shown that AI algorithms can register several features in the images, such as size, homogeneity, gradient, and intensity [5]. This creates an opportunity for a more nuanced analysis than the current binary decision, i.e., mismatch or no-mismatch. 

In the five included studies, we found that the used AI algorithms performed equivalent to or even better than neuroradiologists and neurologists in the binary classification of time since onset ±4.5 h from DWI/FLAIR images. The automated registration of several features could explain these results.

Thomalla et al. and Ebinger et al. have found that even though there is a relationship between the time clock and tissue progression, there is no significant correlation between relative signal intensity on FLAIR imaging and TSS [5,8]. The FLAIR hyperintensity lesions can occur before 4.5 h but may also occur later [5,6,8]. The automated analysis of DWI/FLAIR lesion mismatch could be useful assistance in the daily acute setting concerning lesion visibility, but the determination of TSS from these identified lesions should be considered with caution.

The sensitivities and specificities reported for the AI algorithms were more consistent across the different studies than for the neuroradiologists and neurologists, with the exception of one study, showing the opposite. This supports the idea that automated segmentations could provide a standardized assessment of lesions and potentially decrease inter-rater variability. In a study on fully automatic acute ischemic lesion segmentation on DWI, they argue that domain experts could save time and effort if the manual segmentations were based on automated segmentation [28].

In a study from 2022 on automated segmentation, they tested whether the combination of T2w-FLAIR and DWI sequences could lead to a better segmentation of stroke lesions than single modal modality approaches, i.e., only DWI [20]. Jiang et al. also compared the performance of their model with different modalities, i.e., DWI vs. FLAIR vs. DWI + FLAIR. They found that DWI + FLAIR had the best performance [26].

Similarly, in a study on infarct segmentation on DWI, ADC, and low b-value-weighted images from 2019, they found that the combination of sequences led to an improved segmentation of lesions compared to only diffusion maps and could produce results comparable with manual lesions segmented by domain experts [29]. 

These studies suggest that automated segmentations could assist neuroradiologists in the assessment of infarction lesions and maybe neurologists in the treatment decision of patients with acute ischemic stroke.

### 4.2. FLAIR Hyperintensity Segmentation

To conduct the FLAIR segmentation, three of the included articles, Zhu et al., Jiang et al., and Lee et al., used the DWI segmentation data and features, such as location, to improve their FLAIR segmentation [22,25,26]. Lee et al. used the apparent diffusion coefficient (ADC) maps.

Other studies on FLAIR segmentation have highlighted the difficulties in segmenting FLAIR lesions in stroke and mixed pathology cases, yielding mediocre DICE values ranging from 0.58 to 0.79 [30,31]. This could indicate that the segmentation of FLAIR lesions is not only difficult for the human eye but also a challenge for an AI algorithm.

To improve FLAIR segmentation, the intensity correction by an algorithm, the use of an automated filter to remove inhomogeneity, and the negative influence of high-intensity regions were tested in a study from 2019. The FLAIR lesion segmentation method showed promising results in a follow-up FLAIR imaging dataset on lesion volume estimations [30].

In a study on FLAIR visibility and 90-day outcome prediction but no DWI/FLAIR mismatch assessment, they found that patients with an early visible FLAIR lesion within 4.5 h and who were given intravenous (IV) thrombolysis had a poorer 90-day outcome than those who had a later development of visible FLAIR lesions [32].

Even though FLAIR lesion visibility cannot serve as a surrogate marker for TSS, FLAIR changes could resemble a biological marker of stroke progression/tissue salvage-ability, though the influence of the blood-brain barrier (BBB) permeability and vascular recanalization in subacute ischemic stroke is still questioned [5,8,9,33].

### 4.3. Limitations of the Included Studies

The risks of bias and concern for applicability seen in the four domains of the QUADAS-2 assessment were mainly due to a limited description of the human comparator process, convenience sampled cohorts, insufficient description of inclusion and exclusion criteria, and/or extensive exclusion of images which could limit the use in a daily clinical setting.

Only two of the included studies transparently specified their stroke patient selection with inclusion and exclusion criteria. The exclusion criteria were small or unsegmented lesions, DWI lesions on sites with extensive leucoaraiosis on FLAIR, and heavy artifacts on DWI or FLAIR sequences [25,26]. These exclusion criteria narrow the number of patients suited for the AI algorithms. One study only reported characteristics for the cohort in general, which could raise the risk of an unbalanced training and test cohort selection [22]. Furthermore, another of the five studies did not report the percentage of patients who had a stroke within 4.5 h [23]. This could potentially bias the test data and facilitate better test results, which could introduce an overestimation of model performance and limit integration into clinical practices.

The split size of the training set and test set in all five studies followed the general trend and ranged from 70/30 to 90/10. Consideration of sample size is important as deep-learning-based algorithms have higher recognition accuracy on larger sample data sets [34].

### 4.4. Limitations of This Review

The search process was performed according to PRISMA guidelines. Given the speed and interest to develop AI algorithms for neuroradiology purposes, there is a probability that newer and relevant studies are not included in this systematic review when published.

One of the inclusion criteria was to only include algorithms on DWI and FLAIR sequences, based on the present process to determine treatment eligibility for patients with unknown symptom onset. In the preliminary search process, we found a study on stroke onset determination with the use of perfusion-weighted imaging [35]. This shows the limitation of our search due to the narrow inclusion criteria. Another limitation of this review is the risk of publication bias. It is not possible to rule out the risk of conducted studies with poor performance not being published. Unpublished work with poor performance would lead to a false view of the potential and performance of AI algorithms for mismatch assessment. The five reviewed methods all focus on TSS from the classification of DWI/FLAIR mismatch. We searched for studies on DWI/FLAIR mismatch assessment without excluding articles with a focus on the prediction of outcomes but did not find any that fit the scope of this review. We did find studies on DWI and FLAIR segmentation, which focused on the outcome, but there were no studies on DWI/FLAIR mismatch, outcome prediction, and AI.

### 4.5. Perspectives

The continued advancement of radiology AI methods offers new opportunities to improve the detection, diagnosis, and prediction of patient outcomes. Still, even if the results of a study seem promising, a method can sometimes perform poorer than expected when tested on an external dataset with a larger, consecutive cohort. External validation is, therefore, important, and a validation and evaluation framework could be necessary [36].

For remote hospitals and in countries where the MRI scanner density is low, the use of an approved AI algorithm could assist regarding diagnosis, decrease unnecessary transportation to other clinics, and improve treatment decisions and, thereby, patient outcomes [37].

Time is of the essence in the treatment of acute ischemic stroke. A rapid and efficient workflow is, therefore, essential. The exclusion of hemorrhage, stroke mimics, and the estimation of stroke volume on imaging is important to identify patients eligible for rtPA-treatment [38]. The door-to-needle (DTN) time, which is the time from the arrival of the acute ischemic stroke (AIS) patient at the emergency room to the initiation of rtPA-treatment, can be used as a measurement of quality improvement [39]. An automated stroke detection with a reliable imaging method could possibly decrease DTN time [39]. Another method to improve stroke care is discussed in a study from 2021 on the development of a prediction model for daily stroke occurrences. They describe how the prediction could be improved by synchronizing a variety of medical information and have patients perform self-care. Further knowledge of stroke occurrence could also have an influence on the improvement in medical resources [40]. This could lower DTN time, which would improve stroke care and patient outcome.

Another way to improve MRI workflow was tested in a study on a synthetic FLAIR sequence automatically developed from DWI sequences. The synthetic FLAIR sequence was tested against the usual FLAIR sequence in DWI/FLAIR mismatch assessment by neuroradiologists. It showed comparable diagnostic performances [41].

A number of AI algorithms for outcome prediction of stroke patients based on MRI have used modified Rankin scales as a reference to predict stroke outcomes and were able to predict the likelihood of hemorrhagic transformation [42,43,44,45].

In a Meta-analysis on examination of delayed thrombolysis among patients selected according to mismatch criteria, they found that patients who received delayed thrombolysis, i.e., after 3 h, was associated with increased reperfusion/recanalization, but the relation between delayed thrombolysis in mismatch patients and the post-treatment outcome was not established [46]. Several other studies have later shown the importance of early management, which is also included in the guidelines from 2019 by Power et al. [1].

In WAKE-UP, a randomized, placebo-controlled trial of thrombolysis in stroke with an unknown time of symptom onset, they used MRI criteria to determine patients’ eligibility for thrombolysis. In a report from the WAKE–UP study on systematic image interpretation, the physicians went through systematic image training before rating the images. The results showed a high level of consistency amongst the raters in the assessment of the different imaging criteria, e.g., the presence of an acute ischemic lesion and the extent of infarction [47]. The DWI/FLAIR mismatch assessment was described as the most difficult imaging criteria to rate. The interrater agreement showed good consistency, but it was lower than for the other imaging criteria [47]. Several other studies argue that there is a noticeable interrater variability in the DWI/FLAIR mismatch assessment [6,7,8]. A study from 2018 found an insufficient interrater agreement on mismatch assessment and expressed uncertainty about this method to be the center of decisions in patients with unknown TSS [48]. This emphasizes that the use of an automated assessment could assist the interpretation and potentially improve the current standard.

Adding stroke treatment and patient-specific data, such as the location of lesions, symptoms, and comorbidities, will improve the interpretation of the relationship between imaging features and functional outcomes. A reliable algorithm to serve as assistance to determine patient eligibility for rtPA treatment could prove to provide a more standardized evaluation, be less time-consuming, and overall improve the treatment of patients admitted to the hospital under suspicion of stroke.

However, the creation of an AI method that will lower the DTN time, be accurate, and also bring predictive knowledge of the influenced tissues still comprise a challenge [49].

In all of the reviewed studies and in the daily clinical practice, some patients are misclassified in regard to the cut-off time of 4.5 h. In Polson et al., the DL model identified 29/37 (78%), and the radiologists identified 23/37 (62%) of the evaluation set patients with known onset time within 4.5 h [24]. The AI model used imaging features not only from the lesion area but also from the surrounding brain regions. The results could underline the relevance of the tissue status not only in the stroke lesion but also in the surrounding regions.

To be able to improve the analysis of the tissue status and stroke progression and identify those cases currently misclassified, further research is required. The development of AI methods and awareness in daily clinical practice could be amplified through new research in the field with a status of the currently existing AI methods. R. Karthik and R. Menaka et al. propose the development of an end-to-end automatic framework to identify both stroke lesions and prediction of the outcome of the influenced tissues to increase awareness of existing computer-aided detection frameworks [49].

This review elucidates the existing AI methods on DWI/FLAIR mismatch assessment, creates awareness, and focuses on the next potential steps towards an improvement in the current standard analysis and treatment selection for rtPA.

## 5. Conclusions

In a binary classification of ±4.5 h, the machine learning and deep learning algorithms in this review performed equivalent to or even better than domain experts, but surveyed AI methods are most likely not able to determine TSS only using DWI and FLAIR sequences. Furthermore, the relation between time since stroke onset and increasing visibility of FLAIR hyperintensity lesions is not recommended for the determination of TSS within the first 4.5 h. An AI DWI/FLAIR mismatch assessment focused on treatment eligibility and outcome prediction, adding patient-specific data, could pave the road for improving stroke treatments.

## Figures and Tables

**Figure 1 diagnostics-13-02111-f001:**
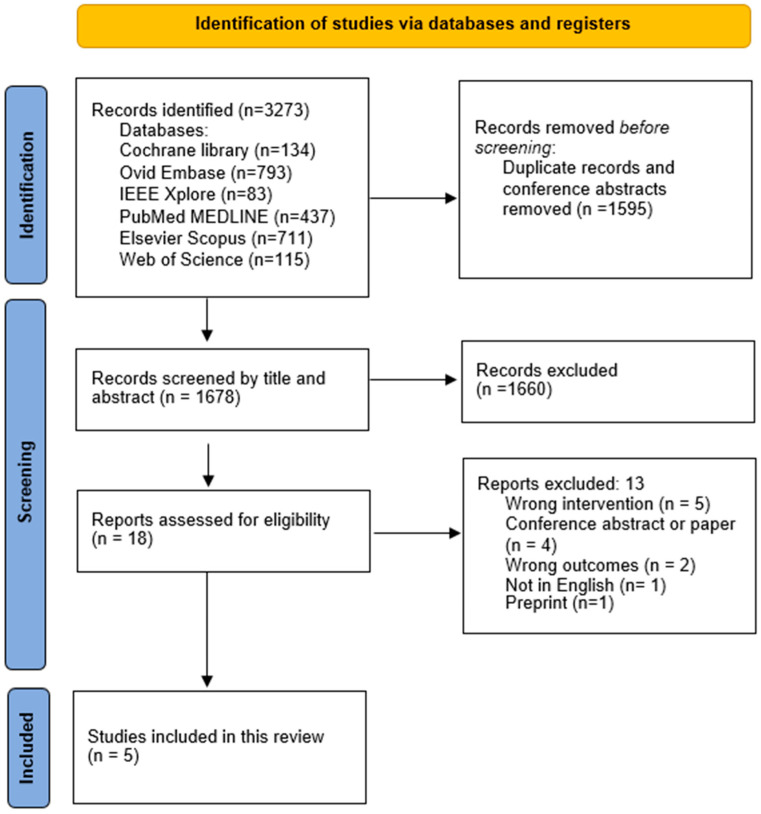
PRISMA workflow. The study inclusion process.

**Table 1 diagnostics-13-02111-t001:** Main in- and exclusion criteria.

Inclusion Criteria	Exclusion Criteria
Machine-learning methods that automatically identify DWI and T2 FLAIR lesions on structural brain MRI. Machine-learning methods with automated DWI/FLAIR mismatch assessment.	Machine learning methods using other specialized MR sequences or modalities than DWI and T2 FLAIR. Machine learning methods with a primary focus on CT.
Machine-learning methods that automatically determine the time since onset of stroke from DWI and FLAIR lesion mismatch.	Insufficient reporting of data acquisition, test strategy, or performance metrics.

**Table 2 diagnostics-13-02111-t002:** Study characteristics and scanner specifications.

Author	N (Training/Test Set (%))	Age (Training Set)	Age (Test Set)	NIHSS at Admission (Training Set)	NIHSS at Admission (Test Set)	Time from Stroke Onset to MRI (Training Set) (Hours)	Strokes within 4.5 h (Training Set) (%)	Strokes within 4.5 h (Test Set) (%)	Scanner Characteristics	Exclusions of Scans Based on Quality Assessment?	AI Algorithms Used for Stroke Segmentation and TSS Classification
		(Years)	(Years)								
Zhu et al., 2021 [22]	161/107 (60.8%/39.2%)	67.7 ¹		9.7 ¹		10.7 ¹	64.6		Multiple 3T of same type	N/A	Segmentation: EfficientNet-B0 TSS classification: Ensemble of SVM, LR, RF, GBDT, ET
Zhang et al., 2020 [23]	340/82 (80.6%/19.4%)	70	68	8	6.5	3.5	N/A	N/A	N/A	Yes	Segmentation and TSS classification: Unified multistage CNN
Polson et al., 2022 [24]	340/74 (82.1%/17.9%)	70	68	8	6.5	3.5	54	50	N/A	Yes	Segmentation and TSS classification: Unified multistage CNN
Lee et al., 2020 [25]	299/54 (84.2%/15.8%)	63	67	4	5	4.5	49.8	61.1	Multiple 1.5T of different brands	Yes	Segmentation: Feature extraction based on Size, intensity, gradient, GLCM, GLRLM, and LBP TSS classification: LR, RF, SVM (linear and radial basis function kernel)
Jiang et al., 2022 [26]	410/177 (69.9%/30.1%)	67.2 ¹	68.8 ¹	11.7 ¹	13.4 ¹	3.52 ¹	60	54.2	Multiple 3T of different brands	Yes	Segmentation: Feature extraction using pyradiomics software TSS classification: RF, Bayes, NN, KNN, Adaboost, SVM

Abbreviations. NIHSS = NIH Stroke Scale, ML = machine learning, DL = deep learning, AdaBoost = adaptive boosting, CNN = convolutional neural network, ET = extra trees, GBDT = gradient boosted decision tree, GLCM = gray-level co-occurrence matrices, GLRLM = gray-level run-length matrices, KNN = K- nearest neighbor, LBP = local binary patterns, LR = logistic regression, NN = neural network, RF = random forest, SVM = support vector machine. All numbered data is a median unless otherwise specified. ¹ signifies the reported data is an average.

**Table 3 diagnostics-13-02111-t003:** Performance results of TSS classification.

Author	AI Algorithms with Performance Reported Here	Testing Strategy	AUC	Sens	Spec	PPV	NPV	Acc	Comparator	Comparator AUC	Comparator Sens	Comparator Spec	Comparator PPV	Comparator NPV	Comparator Acc
Zhu et al., 2021 [22]	Ensemble of SVM, LR, RF, GBDT, ET	Train-test split incl. external test set	0.82	0.77	0.84	0.91 *	0.65 *	0.81	Radiologists mismatch assessment	N/A	0.82	0.59	0.72 *	0.72 *	0.70
Zhang et al., 2020 [23]	CNN	Train-test split	0.74	0.70	0.81	N/A	N/A	0.76	Radiologists mismatch assessment	N/A	0.57	0.88	N/A	N/A	0.72
Polson et al., 2022 [24]	CNN	Train-test split incl. external test set	0.77	0.71	0.74	0.74 *	0.91 *	0.73	Radiologists mismatch	N/A	0.62	0.86	0.86 *	0.62 *	0.74
Lee et al., 2020 [25]	RF	Train-test split	0.85	0.76	0.83	0.86	0.70	0.79 *	Stroke Neurologists mismatch assessment	N/A	0.49	0.91	0.89	0.55	0.61 *
Jiang et al., 2022 [26]	SVM (Radial)	Train-test split incl. external test set	0.90	0.82	0.83	0.85	0.80	0.83	N/A	N/A	N/A	N/A	N/A	N/A	N/A

Only the best performance metrics preferentially from out-of-distribution external validation datasets are reported in this table. * signifies the values are calculated. Abbreviations: AUC = area under receiver operatic characteristics curve; ACC = accuracy, CNN = convolutional neural network, ET = extra trees, GBDT = gradient boosted decision tree, LR = logistic regression, RF = random forest, SVM(Radial) = support vector machine with radial basis function kernel, Sens = sensitivity; Spec = specificity; PPV = positive predictive value; NPV = negative predictive value. All numbered data is a median unless otherwise specified.

**Table 4 diagnostics-13-02111-t004:** Tabular presentation of the QUADAS-2 assessments in each study.

Source	Risk of Bias	Concern for Applicability
PatientSelection	Index Test	Reference Standard	Flow and Timing	PatientSelection	Index Test	Reference Standard
Zhu et al. [22]	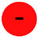		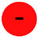	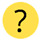	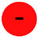		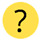
Zhang et al. [23]	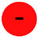			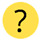	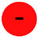		
Polson et al. [24]					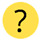	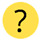	
Lee et al. [25]					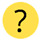		
Jiang et al. [26]	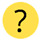			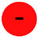	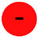		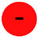


 = low risk of bias and concern for applicability, 
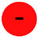
 = high risk of bias and concern for applicability, 
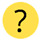
 = unclear risk of bias and concern for applicability.

## Data Availability

Data are available upon reasonable request.

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
