# Peer review of "Artificial Intelligence for Automated DWI/FLAIR Mismatch Assessment on Magnetic Resonance Imaging in Stroke: A Systematic Review"

_diagnostics, 2023, doi:10.3390/diagnostics13122111_

Round 1
Reviewer 1 Report
Authors need to address the following suggestions.
1. Only 5 articles were analyzed in this review. Authors need to double check once if any other manuscript is missed in the review process.
2. Many review articles were presented for the detection of ischemic stroke. Authors need to discuss those work and clearly present, what are the new findings in this review. Following are some of those works that need to be cited and discussed.
Imaging assessment of acute ischaemic stroke: a review of radiological methods
Computer-aided detection and characterization of stroke lesion–a short review on the current state-of-the art methods
3. Discussion section need to be strengthened.
Minor edits are required
Author Response
Response to Reviewer 1 Comments
Dear Reviewer 1
Thank you for your review of our manuscript and the comments and suggestions you provide. In the following, we will reply to each of your numbered comments.
Point 1: Only 5 articles were analyzed in this review. Authors need to double check once if any other manuscript is missed in the review process.
Response 1: Thank you for your comment. Yes. Only five articles were analyzed in this review. We have performed another search for the given time-period in the review, and we did not find any manuscripts to fit the scope of the review that was not already included. We have also done a PubMed search covering the period from November 20, 2022 until present day. This search was performed on May 31, 2023. It did not display a new study suitable for inclusion. To the best of our knowledge the five included articles is the correct five articles when considering our inclusion and exclusion criteria. We discuss the specific scope of our study in section 4.4 Limitations of This Review.
2. Many review articles were presented for the detection of ischemic stroke. Authors need to discuss those work and clearly present, what are the new findings in this review.
Following are some of those works that need to be cited and discussed.
Imaging assessment of acute ischaemic stroke: a review of radiological methods
Computer-aided detection and characterization of stroke lesion–a short review on the current state-of-the art methods
Response 2:
Thank you for your suggestion. We have referenced the two studies you mention above in section 4. Discussion.
In section 4.5 Perspectives we have tried to clarify how the results of this review can prove to be relevant for clinicians and how the review of the current methods can bring focus on what the next steps should be to improve the current standard of analysis and treatment.
3. Discussion section need to be strengthened
Response 3:
Thank you for your comment.
We have made revisions in section 4. Discussion.
We have used additional references including the two you suggested and made revisions throughout the section.
We hope that you find our revisions have improved the manuscript satisfactorily.
Yours sincerely,
The authors
--------------------------------------
Reviewer 2 Report
Thank you for your good report on DWI/FLAIR mismatch detection system based on AI. This review has an impact for neurologists and is very helpful for all of the physicians.
Abst:
Align decimal points.
If you are writing a paper on AI, you would probably have Precision and Fvalue as well as Sensitivity Specificity.
Intro; Good
Method;Good
Result; Please describe sensitivity ,specificity, precision, F value of each research.
Discussion;
- PMID: 33598347 Please referre this article and Please state that further time savings could be achieved if the automation of stroke prediction and imaging to decision making could be achieved.
The presence of a mismatch and the administration of t-PA is ultimately left to the neurologist. Are there any reports of real world data where a mismatch was suspected but t-PA was administered and it worked? Also, what is the probability that a mismatch is not found and missed? In addition, I think there is some hesitation in administering t-PA if there is an extensive DWI image of infarction.
Please summarize the neurologist's opinion on how to combine AI suggestions with actual clinical judgment.
Author Response
Response to Reviewer 2 Comments
Thank you for your good report on DWI/FLAIR mismatch detection system based on AI. This review has an impact for neurologists and is very helpful for all of the physicians.
Dear Reviewer 2
Thank you for your review of our manuscript. In the following, we will address your
comments.
Point 1: Abst: Align decimal points.
Response 1: We had one value registered with three decimals to use the value from the reviewed papers, but we do agree that an alignment is suited. The value is now revised and has only two decimal points. We have aligned throughout the paper.
Point 2: If you are writing a paper on AI, you would probably have Precision and F value as well as Sensitivity Specificity.
Response 2: We went through the five papers and only one of them used the term
Precision (also known as positive predictive value) in their results section. We do have positive predictive value either registered or calculated in four out five studies. In section 3. Results we have described that one of the studies use the term Precision, and hope you will find our explanation is to your satisfaction.
For further explanation of our use of Sensitivity, Specificity, and F-value please see
Response 5.
Point 3: Intro; Good
Response 3: Thank you for your comment.
Point 4: Method; Good
Response 4: Thank you for your comment.
Point 5: Result; Please describe sensitivity, specificity, precision, F value of each research.
Response 5: The Sensitivity, also known as true positive rate, and the Specificity, also known as true negative rate, were reported in all five studies. It ranged from 0.70 to 0.82 and from 0.74 to 0.84, respectively. When considering the best performances of the AI methods, the Specificity was a little higher than the Sensitivity in all five studies. The exact values are shown in Table 3, section 3. Results.
None of the five studies describes the use of F-test nor F-value and we have chosen to use the registrations and descriptions from the studies, which is also well suited for these works.
The articles report the composite metric AUC. We have chosen to follow their use, and describe their results through their choice of values.
To align the studies we have calculated PPV and NPV when possible.
Point 6: Discussion; PMID: 33598347 Please referre this article and Please state that further time savings could be achieved if the automation of stroke prediction and imaging to decision making could be achieved.
The presence of a mismatch and the administration of t-PA is ultimately left to the
neurologist.
Response 6: Thank you for your suggestion. We have referenced the study in section 4.5 Perspectives.
Yes, the neurologist will be the one to make the decision on whether the patient should receive thrombolysis. With relevant communication between the neuroradiologist and neurologist, the decision can be improved through their combined expertise.
Point 7: Are there any reports of real world data where a mismatch was suspected but t-PA was administered and it worked?
Also, what is the probability that a mismatch is not found and missed? In addition, I think there is some hesitation in administering t-PA if there is an extensive DWI image of infarction.
Patients diagnosed with acute ischemic stroke (AIS) with magnetic resonance DWI lesions but no visible FLAIR hyperintensity lesions, i.e. mismatch, are likely to be within the treatment window for treatment with thrombolysis [1].
In a study on FLAIR visibility and 90-day outcome prediction, but no DWI/FLAIR mismatch assessment, they found that patients with an early visible FLAIR lesion within 4.5 hours, and who were given intravenous (IV) thrombolysis, had a poorer 90-day outcome than those who had a later development of visible FLAIR lesions [2].
In a Meta-analysis on examination of delayed thrombolysis among patients selected according to mismatch criteria they found that patients who received delayed thrombolysis, i.e. after 3 hours, was associated with increased reperfusion/recanalization, but the relation between delayed thrombolysis in mismatch patients and post-treatment outcome was not established [3].
Several other studies have later shown the importance of early management, which is also included in the guidelines from 2019 by Power et al. [4].
In the WAKE-UP randomized, placebo-controlled trial of thrombolysis in stroke with unknown time of symptom onset, they used magnetic resonance imaging criteria to determine patients' eligibility for thrombolysis. The interrater agreement between the clinicians, who assessed and rated the DWI/FLAIR mismatch, was evaluated. The raters were trained in systematic image assessment. The result showed a high level of consistency amongst raters in the assessment of the various imaging criteria, including the presence of an acute ischemic lesion and the extent of infarction [5].
With the appropriate image training the probability of missing stroke lesions, is most likely low.
It is correct that a large DWI lesion can create some hesitation concerning the
administration of rtPA. In the WAKE-UP trial, a DWI lesion volume > 1/3 of the Middle Cerebral artery (MCA) or > 50% of the Anterior cerebral artery (ACA) or posterior cerebral artery territory (visual inspection) or > 100 mL was an exclusion criteria for thrombolysis.
The raters in the WAKE-UP study described the DWI/FLAIR mismatch as the hardest imaging assessment to do. The interrater agreement showed a good consistency, but it was a little lower than for the other imaging criteria. Several other studies argue that there is a noticeable interrater variability in the DWI/FLAIR mismatch assessment [1, 6, and 7].
In a study from 2018, they found an insufficient interrater agreement on the mismatch assessment and express uncertainty about this method to be the center of decision in patients with unknown stroke onset time [8].
This emphasizes that the use of AI could improve the assessment. It could create a more standardized method, assist the stroke progression analysis and treatment selection process.
Point 8: Please summarize the neurologist's opinion on how to combine AI suggestions with actual clinical judgment.
Response 8: From the neurologists perspective an AI assisted DWI/FLAIR mismatch
assessment could be useful especially in the following situations:
- Support the decision of treatment selection in cases where there is uncertainty regarding the mismatch assessment.
- Support the image analysis in remote locations and other situations where a
neuroradiologist is not present. - To be used in selection in new clinical trials and then potentially be part of future clinical routines for patient selection.
Thank you for the comments and suggestions you provided.
We hope that you agree on our revisions and find that our manuscript is now improved in an acceptable manner.
Yours sincerely,
The Authors
References:
1. Thomalla G, Cheng B, Ebinger M, Hao Q, Tourdias T, Wu O, Kim JS, Breuer L, Singer OC, Warach S, Christensen S, Treszl A, Forkert ND, Galinovic I, Rosenkranz M, Engelhorn T, Köhrmann M, Endres M, Kang DW, Dousset V, Sorensen AG, Liebeskind DS, Fiebach JB, Fiehler J, Gerloff C; STIR and VISTA Imaging Investigators. DWI-FLAIR mismatch for the identification of patients with
acute ischaemic stroke within 4·5 h of symptom onset (PRE-FLAIR): a multicentre observational study. Lancet Neurol. 2011 Nov;10(11):978-86. doi: 10.1016/S1474-4422(11)70192-2. Epub 2011 Oct 4. PMID: 21978972.
2. Kim Y, Luby M, Burkett NS, Norato G, Leigh R, Wright CB, Kern KC, Hsia AW, Lynch JK, Adil MM, Latour LL. Fluid-Attenuated Inversion Recovery Hyperintense Ischemic Stroke Predicts Less Favorable 90-Day Outcome after Intravenous Thrombolysis. Cerebrovasc Dis. 2021;50(6):738- 745. doi: 10.1159/000517241. Epub 2021 Jul 20. PMID: 34284378; PMCID: PMC8639625.
3. Mishra NK, Albers GW, Davis SM, Donnan GA, Furlan AJ, Hacke W, Lees KR. Mismatch-based delayed thrombolysis: a meta-analysis. Stroke. 2010 Jan;41(1):e25-33. doi: 10.1161/STROKEAHA.109.566869. Epub 2009 Nov 19. Erratum in: Stroke. 2010 Apr;41(4):e399. PMID: 19926836.
4. Powers WJ, Rabinstein AA, Ackerson T, Adeoye OM, Bambakidis NC, Becker K, Biller J, Brown M, Demaerschalk BM, Hoh B, Jauch EC, Kidwell CS, Leslie-Mazwi TM, Ovbiagele B, Scott PA, Sheth KN, Southerland AM, Summers DV, Tirschwell DL. Guidelines for the Early Management of Patients With Acute Ischemic Stroke: 2019 Update to the 2018 Guidelines for the Early Management of Acute Ischemic Stroke: A Guideline for Healthcare Professionals From the American Heart Association/American Stroke Association. Stroke. 2019 Dec;50(12):e344-e418. doi: 10.1161/STR.0000000000000211. Epub 2019 Oct 30. Erratum in: Stroke. 2019 Dec;50(12):e440-e441. PMID: 31662037.
5. Galinovic I, Dicken V, Heitz J, Klein J, Puig J, Guibernau J, Kemmling A, Gellissen S, Villringer K, Neeb L, Gregori J, Weiler F, Pedraza S, Thomalla G, Fiehler J, Gerloff C, Fiebach JB; WAKE-UP Investigators. Homogeneous application of imaging criteria in a multicenter trial supported by investigator training: A report from the WAKE-UP study. Eur J Radiol. 2018 Jul;104:115-119. doi: 10.1016/j.ejrad.2018.05.011. Epub 2018 May 15. PMID: 29857856.
6. Thomalla G, Rossbach P, Rosenkranz M, Siemonsen S, Krützelmann A, Fiehler J, Gerloff C. Negative fluid-attenuated inversion recovery imaging identifies acute ischemic stroke at 3 hours or less. Ann Neurol. 2009 Jun;65(6):724-32. doi: 10.1002/ana.21651. PMID: 19557859.
7. Emeriau S, Serre I, Toubas O, Pombourcq F, Oppenheim C, Pierot L. Can diffusion-weighted imaging-fluid-attenuated inversion recovery mismatch (positive diffusion-weighted imaging/negative fluid-attenuated inversion recovery) at 3 Tesla identify patients with stroke at <4.5 hours? Stroke. 2013 Jun;44(6):1647-51. doi: 10.1161/STROKEAHA.113.001001. Epub 2013 May 2. PMID: 23640823.
8. Fahed R, Lecler A, Sabben C, Khoury N, Ducroux C, Chalumeau V, Botta D, Kalsoum E, Boisseau W, Duron L, Cabral D, Koskas P, Benaïssa A, Koulakian H, Obadia M, Maïer B, Weisenburger-Lile D, Lapergue B, Wang A, Redjem H, Ciccio G, Smajda S, Desilles JP, Mazighi M, Ben Maacha M, Akkari I, Zuber K, Blanc R, Raymond J, Piotin M. DWI-ASPECTS (Diffusion-Weighted Imaging- Alberta Stroke Program Early Computed Tomography Scores) and DWI-FLAIR (Diffusion-
Weighted Imaging-Fluid Attenuated Inversion Recovery) Mismatch in Thrombectomy Candidates: An Intrarater and Interrater Agreement Study. Stroke. 2018 Jan;49(1):223-227. doi: 10.1161/STROKEAHA.117.019508. Epub 2017 Nov 30. PMID: 29191851.